# The Stability of the Anti-Müllerian Hormone in Serum and Plasma Samples under Various Preanalytical Conditions

**DOI:** 10.3390/diagnostics13081501

**Published:** 2023-04-21

**Authors:** Radana Vrzáková, Václav Šimánek, Ondřej Topolčan, Vladimír Vurm, David Slouka, Radek Kučera

**Affiliations:** 1Department of Chemistry and Biochemistry, Faculty of Medicine in Pilsen, Charles University, 32300 Pilsen, Czech Republic; radana.vrzakova@lfp.cuni.cz; 2Department of Immunochemistry, University Hospital and Faculty of Medicine in Pilsen, Charles University, 32300 Pilsen, Czech Republic; simanek@fnplzen.cz (V.Š.); topolcan@fnplzen.cz (O.T.); lekar@centrum.cz (V.V.); kucerar@fnplzen.cz (R.K.); 3Department of Otorhinolaryngology, University Hospital and Faculty of Medicine in Pilsen, Charles University, 30955 Pilsen, Czech Republic; 4Department of Pharmacology and Toxicology, Faculty of Medicine in Pilsen, Charles University, 32300 Pilsen, Czech Republic

**Keywords:** anti-Müllerian hormone (AMH), serum, plasma, preanalytical conditions, stability

## Abstract

The anti-Müllerian hormone (AMH) is a glycoprotein that plays an important role in prenatal sex differentiation. It is used as a biomarker in polycystic ovary syndrome (PCOS) diagnostics, as well as for estimating an individual’s ovarian reserve and the ovarian response to hormonal stimulation during in vitro fertilization (IVF). The aim of this study was to test the stability of AMH during various preanalytical conditions that are in accordance with the ISBER (International Society for Biological and Environmental Repositories) protocol. Plasma and serum samples were taken from each of the 26 participants. The samples were then processed according to the ISBER protocol. AMH levels were measured in all the samples simultaneously using the chemiluminescent kit ACCESS AMH in a UniCel^®^ DxI 800 Immunoassay System (Beckman Coulter, Brea, CA, USA). The study proved that AMH retains a relatively high degree of stability during repeated freezing and thawing in serum. AMH was shown to be less stable in plasma samples. Room temperature proved to be the least suitable condition for the storage of samples before performing the biomarker analysis. During the testing of storage stability at 5–7 °C, the values decreased over time for all the plasma samples but remained stable in the serum samples. We proved that AMH is highly stable under various stress conditions. The anti-Müllerian hormone retained the greatest stability in the serum samples.

## 1. Introduction

AMH is a glycoprotein dimer consisting of two monomers with a molecular weight of 72 kDa each [1]. AMH belongs to the group of transforming growth factors-β [2]. The gene for AMH is located on the short arm of chromosome 19 [3]. AMH acts on the cells of the gonads and the Müllerian ducts via two types of receptors: MIS receptor type I (MISRI) and MIS receptor type II (MISRII) [4]. AMH is formed in the Sertoli cells of the testes prenatally as well as postnatally. In women, the production of AMH occurs postnatally in the granulosa and theca cells of the ovaries [5,6].

AMH plays a role in embryo gender differentiation [1,7]. It induces the regression of the Müllerian ducts, while testosterone secreted from Leydig cells induces the development of the Wolffian ducts from which the internal male genitalia are formed [1,8]. The absence of AMH in the female fetus causes the development of Müllerian ducts and the absence of androgens leads to the regression of the Wolffian ducts, resulting in the formation of female genitals [8,9].

In men, the level of AMH remains high from birth to puberty and then decreases. In adult men, AMH levels are very low. The level of AMH in girls is very low after birth, but its production gradually increases from around their twelfth year of life onwards. AMH production peaks between the ages of 25–27 years in women and subsequently decreases as the number of follicles decrease [2]. It is undetectable during menopause [10].

AMH is used to estimate the individual ovarian egg reserve of a patient and the biological age of their ovaries [11,12]. Furthermore, it is used to estimate the ovarian response to hormonal stimulation during IVF [13,14]. It is also used in the diagnosis of PCOS [8,15,16] and as a sensitive tumor marker suitable for the diagnosis and monitoring of ovarian granulosa cell tumors (GCT). Serum levels of AMH are elevated in women affected by both PCOS and GCT. AMH is a specific marker in the follow-up of patients who have undergone ovariectomy for GCT, enabling the early detection of recurrences [3]. It is a good predictor of the damage caused to the ovarian follicles during radiotherapy and chemotherapy [8,10,17]. AMH levels are the basis for choosing the right infertility treatment strategy, especially in gonadotropin stimulation during IVF [18,19].

AMH is primarily used in gynecology, but it can also prove to be of use in pediatrics, namely in the diagnosis of sexual development differentiation disorders in children. The production of AMH is lower in prepubertal boys with cryptorchidism than normal boys. This is probably the result of fewer Sertoli cells in the cryptorchid gonads, or a reduced capability of the Sertoli cells to secrete AMH [20]. The absence of AMH is predictive of absent testicular tissue (anorchia) [21].

The International Society of Biological and Environmental Repositories is the leading institution in global biobanking and a pillar of the biorepository community. Its main objective includes monitoring the quality of samples. ISBER focuses on creating guidelines and protocols for the preanalytical and storage conditions of biological samples; the institution has drawn on its own rich experience in their creation [22].

The present study aims to determine the stability of AMH in biological material stored for diagnostic and scientific purposes. The design of the study was prepared according to the ISBER standard operating procedures for serum or plasma sample stability testing.

The aim of this study was to test the stability of AMH in two types of samples during various preanalytical conditions (repeated thawing, different temperature, and storage time) that are in accordance with the ISBER protocol.

## 2. Materials and Methods

### 2.1. Group of Patients

Our cohort consisted of 26 volunteers (2 male and 24 female medical students), whose mean age was 22 years (21–23 years). The minimum number of participants required for significance was calculated using a sample size analysis (detailed description in Section 2.4. Statistical analysis). In female subjects, the samples were collected independently of the day of their menstrual cycle. None of the subjects reported being treated for, or having a history of, endocrinological or fertility disorders. Subjects were not excluded based on their use of hormonal contraception (HC).

### 2.2. Blood Samples

Peripheral blood was drawn using VACUETTE^®^ Z Serum Sep tubes and VACUETTE^®^ lithium heparin (LH) Plasma Sep tubes (Greiner Bio-One, Kremsmünster, Austria). Both serum and plasma samples were collected from each participant using a closed tube Vacuette system. The samples were centrifuged, and serum and plasma separated within 1 h from the blood collection. The range of AMH values in women was between 2.10 and 8.91 ng/mL, and 14.76–17.74 ng/mL in men.

The samples were processed according to the ISBER protocol—the Standard Operating Procedure (SOP) for sample stability testing [23]. All samples were divided into 19 aliquots with an equal sample volume. It is important that each aliquot contains the same sample volume and that the same kind of reaction test tube is used, since unequal sample volumes may affect the concentration of the biomarker due to absorption or evaporation [23]. As a first step, the level of AMH was measured in the fresh aliquots before they were frozen. The following procedure was used to test the freeze–thaw stability of the biomarker. The first six aliquots were frozen at −80 °C. Aliquot 1 was kept at −80 °C throughout. Subsequently, after 12 h, aliquots 2–6 were thawed for two hours at room temperature and then were frozen again at −80 °C. After a subsequent 12-h period, aliquots 3–6 were thawed and frozen the same way and stored at −80 °C. After yet another 12-h period, aliquots 4–6 were thawed, frozen again and stored at −80 °C. Then, 12 h after that, aliquots 5–6 were thawed and again frozen and finally, only aliquot 6 was thawed and frozen again. All aliquots were then stored at −80 °C.

The storage stability test followed. At time t = 0 h, aliquots 7–12 were stored at room temperature and another six aliquots, 13–18, were stored in the fridge at 4 °C. At time points t = 1 h, t = 2 h, t = 4 h, t = 24 h, t = 72 h, and t = 168 h, one sample stored at each temperature (RT and 4 °C) was transferred to −80 °C. Aliquot 19 was stored at −20 °C for one month and then it was placed at −80 °C. Finally, all the samples were stored in the freezer at −80 °C (Table 1). One week after the storage of the last sample at −80 °C, all the samples were thawed and the level of AMH was measured in all the samples at once.

### 2.3. Sample Analysis

The AMH levels were measured using the Access AMH chemiluminescent assay (Beckman Coulter, Brea, CA, USA) with a Limit of Detection (LoD) of ≤0.02 ng/mL and a Limit of Quantitation (LoQ) of ≤0.08 ng/mL. The assay is a simultaneous one-step sandwich immunoassay, which utilizes two monoclonal mouse antibodies for the recognition of total AMH [24]: the capture antibody F2B/12H was coated with paramagnetic particles and the detection antibody F2B/7A was conjugated with alkaline phosphatase [25]. All measurements were performed using the UniCel^®^ DxI 800 Immunoassay System (Beckman Coulter, Brea, CA, USA).

### 2.4. Statistical Analysis

Statistical data analysis was performed using SAS software (SAS Institute Inc., Cary, NC, USA). All graphs were performed using SW STATISTICA (StatSoft, Inc., Tulsa, OK, USA).

The sample size analysis was performed first. The minimum number of participants required for significance was calculated as 21 (females only), based on a test of equivalence with a +/− 0.30 ng/mL limit of tolerance and a test power of 90%.

The mean value, standard deviation (SD), variance, median, interquartile range, minimum, and maximum were calculated for the measured parameters.

An equivalence test (paired design) using two one-sided tests has been used to assess the equivalence of AMH between the initial and the tested value.

A *p*-value of less than 0.05 was considered to be statistically significant.

## 3. Results

AMH levels differ between men and women [26]. At the age of 22, men exhibit significantly higher AMH values than women [10]. This was reflected in our study. The samples taken from our male subjects showed significantly higher levels. However, given that our focus is mainly the role of AMH in IVF, we decided to exclude the male samples after the preliminary statistical analysis.

The difference in the distribution of initial AMH values in serum and plasma were statistically insignificant (*p* = 0.8858) (Figure 1).

Serum samples showed no change in the stability of AMH during freezing and thawing, whereas AMH values in the plasma samples decreased over time (Figure 2). Equivalence was found using ±0.30 ng/mL as the limit of tolerance, but only for serum samples (*p* = 0.0196) and not for plasma (*p* = 0.3778).

The stability of AMH stored at RT after separation decreased for both types of samples over time (Figure 3). An equivalence test found no equivalence for either serum or plasma when using a ±0.30 ng/mL limit of tolerance (*p* = 0.9790 for serum, *p* = 1.0000 for plasma).

The stability of AMH measured in the serum samples stored at 4 °C after separation remained unchanged; meanwhile, the level of AMH in the plasma samples stored at the same temperature decreased over time (Figure 4), the equivalence was proved at the limit of tolerance +/− 0.30 ng/mL only for serum (*p* = 0.0431), not for plasma (*p* = 0.3487).

The stability of AMH measured in the samples stored for 30 days after separation at −20 °C did not change significantly. Neither type of sample showed a change in the concentration of AMH over time (Figure 5), and the equivalence was proved at the limit of tolerance ±0.30 ng/mL (*p* < 0.0001 for serum, *p* = 0.0238 for plasma).

## 4. Discussion

The preanalytical phase is an important component of high-quality output in any laboratory. In recent years, considerable attention has been given to the effect of preanalytical conditions on laboratory results [27]. Millions of clinical samples are obtained every day for use in diagnostic tests that support clinical decision-making. Sample quality must be standardized as a basic condition that ensures accurate results in collected samples [28]. Only appropriate biomarkers measured in samples of adequate quality can improve the diagnostic process and the clinical management of patients [29].

AMH may circulate in the blood in different isoforms. It is produced as a preproprotein that is turned into a proprotein, pro-AMH, after its first cleavage. A second cleavage forms a noncovalent amino and carboxy—terminal dimer (AMHN, AMHC) [30]. It has been reported that the blood of both men and women contains significant levels of pro-AMH, AMHN, and AMHC under physiological conditions. AMHC is the receptor-activating part of this dimer [31]. AMH is involved in the regulation of follicle growth. This study was performed using a commercial assay with defined binding characteristics of the couple of antibodies used [25]. The results of our study must therefore always be assessed with regard to the characteristics of the antibodies used in the assay.

PCOS is associated with an increased presence of the active AMHN isoform. It is AMHN which may be responsible for the anovulation associated with PCOS. Metabolic changes are implicated in this impaired ratio of active/inactive AMH isoforms: increased Body Mass Index (BMI), increased level of C-peptide, increased level of low-density lipoprotein cholesterol and triglycerides, etc. [31].

While it had generally been assumed that AMH is released into the bloodstream at a certain constant rate throughout the menstrual cycle, with minimal intra-individual variability [32], recent publications reported fluctuations in AMH levels during the menstrual cycle [33,34,35]. Such fluctuations can cause various clinical issues when assessing the ovarian reserve or administering gonadotropin. In our study, we focus on determining the stability of AMH released into the bloodstream at the time of sample collection and therefore the methodology of our study is not affected by these facts. On the contrary, since the biomarker is more variable than had been believed, the stability of the biomarker in preanalytical conditions can be perceived as good news for clinicians.

AMH levels can decrease in women as a result of HC administration [36,37,38]. As our focus is the blood concentration of AMH at the time of sample collection, this dip in AMH levels does not present a problem in our study.

To summarize, AMH is a commonly used biomarker in clinical practice, and important decisions are made based on its measurement. Unfortunately, our knowledge of its biological variability and familiarity with all its isoforms and their individual functions is not complete. Studies on the stability of the AMH molecule are therefore very important, even if carried out with the above-mentioned limitations.

The stability of circulating biomarkers can be affected by repeated freezing and thawing [39]. The first point addressed in the ISBER stability protocol is the effect of repeated freezing and thawing on the plasma and serum concentrations of a given analyte. In our study, the results show a relatively high stability of AMH during repeated freezing and thawing for all the serum samples. A decrease in the AMH level was demonstrated in the plasma samples after each cycle (Figure 2).

Knowledge of the effects of temperature and storage time on any given sample is a part of correct laboratory practice [40]. The second half of the ISBER stability protocol is focused on the length of storage time and different temperature conditions. The samples were stored at various temperatures and for various lengths of time. Room temperature proved to be the least suitable for storing samples before an analysis is performed; a statistically significant decrease in AMH was shown in both serum and plasma samples after they were stored at room temperature (Figure 3). When testing the storage stability at 4 °C, the values decreased in the plasma samples over time, but not in the serum samples (Figure 4).

The short-term storage (30 days) of samples at −20 °C resulted in the AMH molecule retaining high stability in both serum and plasma samples (Figure 5).

Our study focused on the stability of AMH after blood was separated. However, the stability of AMH in whole blood samples has been described in the literature. Fleming et al. [41] reported a significant increase in AMH levels in whole blood samples that were stored in the laboratory (20 h, 44 h, and 90 h) at room temperature (20 °C), as well as a slight increase in AMH levels in whole blood samples stored for 90 h in the fridge (4 °C).

It is evident that the cause of the increase in AMH is the fact that the blood was not separated, which allowed for proteolytic enzymes contained in the cellular components of the whole blood to cleave the AMH molecule without the AMH decreasing. This resulted in a paradoxical increase in AMH concentrations. In separated blood samples on the other hand, the level of AMH remained stable even during prolonged storage (5 days at 20 °C and then 2 days at 4 °C).

It is important to know the conditions under which the AMH molecule remains stable. Many samples are transported from smaller laboratories or collection points to a central laboratory; this is a process which requires strict protocols for sampling and further handling [42]. It is common to transport samples after centrifugation, resulting in the transportation of either serum or plasma [43]. Kumar et al. tested the stability of AMH in aliquots of 10 serum samples and 10 LH plasma samples during different storage and freeze–thaw conditions. They showed that AMH maintains its stability in serum samples better than it does in plasma samples under different storage conditions [44]. Different studies have devised different ways to test the stability of the biomarker under various, non-standardized conditions. In our study, we emphasize testing under specific conditions according to the SOPs recommended by the ISBER for testing sample stability [23]. These SOPs very effectively combine tests for the various conditions that are most likely to occur during the transport of the sample.

Our study showed that the decrease in AMH concentration over time in the absence of suitable preanalytical conditions may lead to the measurement of low concentrations of AMH that do not reflect reality. The precise measurement of AMH levels is one of the most important factors for estimating the ovarian response to IVF and choosing the right protocol—the basis of successful fertilization [14,45]. A result showing false low AMH levels may result in the excessive administration of gonadotropins to stimulate the ovaries, which can lead to hyperstimulation syndrome and its severe complications [46].

## 5. Conclusions

The presented study proved that AMH can remain very stable under various stress conditions. The biomarker was most stable in serum samples. Failure to comply with the necessary storage conditions may lead to a decrease in the concentration of AMH in a sample, resulting in a falsely measured low value and this may have clinical consequences, especially during IVF.

## Figures and Tables

**Figure 1 diagnostics-13-01501-f001:**
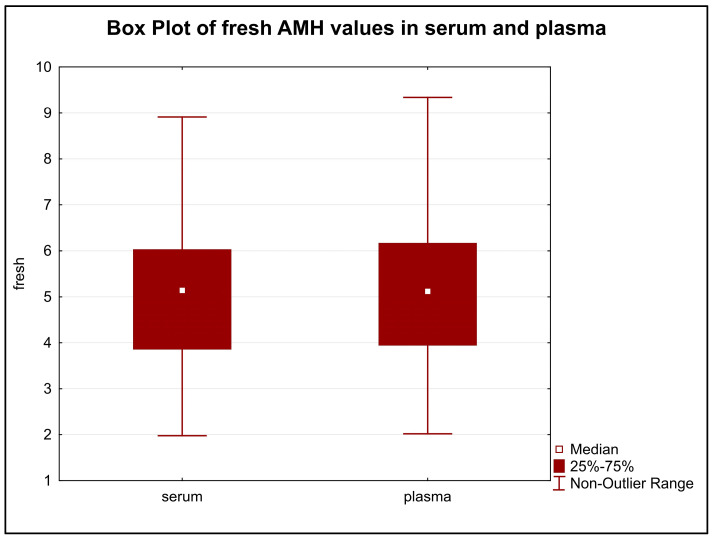
The difference in the distribution of initial AMH values in serum and plasma. Legend: The distribution of the values in serum and plasma samples were very similar and showed no statistically significant difference (*p* = 0.8858).

**Figure 2 diagnostics-13-01501-f002:**
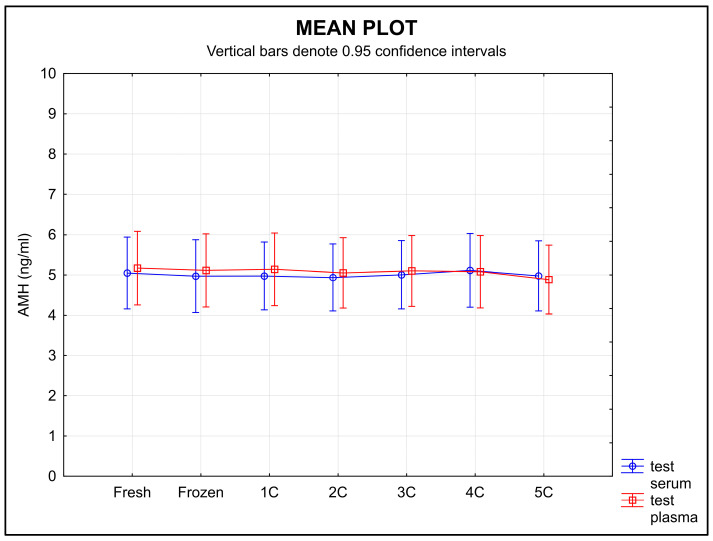
AMH stability during freezing in serum and plasma samples. Legend: The stability of the samples during freezing and thawing did not change over time in serum samples, whereas the stability of the plasma samples decreased over time. An equivalence test using a ±0.30 ng/mL limit of tolerance found equivalence only for serum (*p* = 0.0196), but not for plasma (*p* = 0.3778); 1C, 2C, 3C, 4C, 5C-number of freeze–thaw cycles.

**Figure 3 diagnostics-13-01501-f003:**
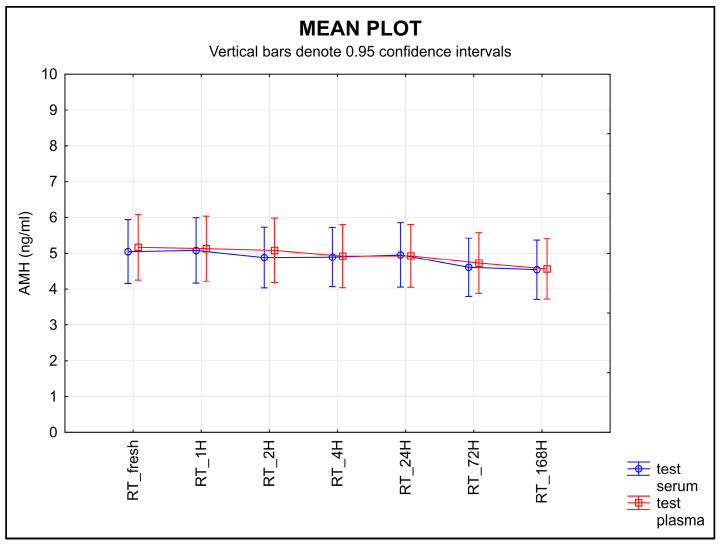
The stability of AMH after separation, stored at RT. Legend: The stability of AMH after separation, stored at RT, decreased for both types of samples with time. An equivalence test using a ±0.30 ng/mL limit of tolerance failed to find equivalence for either serum or plasma samples (*p* = 0.9790 for serum, *p* = 1.0000 for plasma).

**Figure 4 diagnostics-13-01501-f004:**
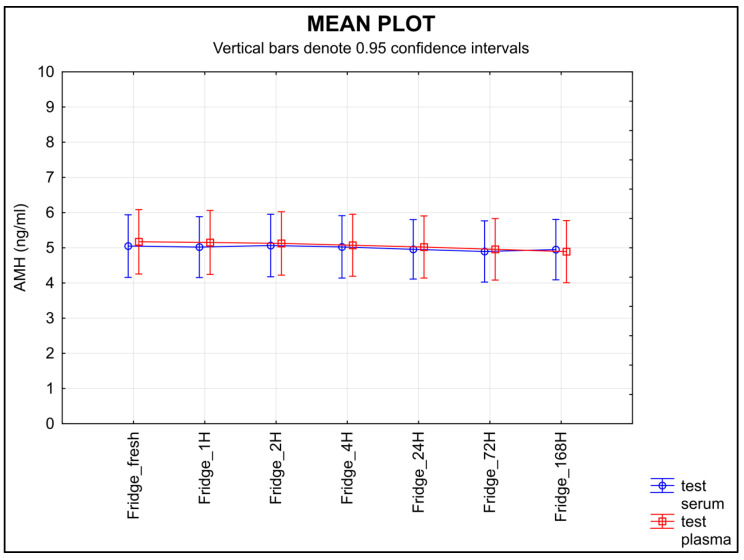
The stability of AMH after separation, stored at 4 °C. Legend: The stability of AMH measured in the samples stored at 4 °C after separation remained unchanged in the serum samples, whereas the level of AMH in the plasma samples decreased over time. An equivalence test using a ±0.30 ng/mL limit of tolerance found equivalence only for serum samples (*p* = 0.0431), but not for plasma (*p* = 0.3487).

**Figure 5 diagnostics-13-01501-f005:**
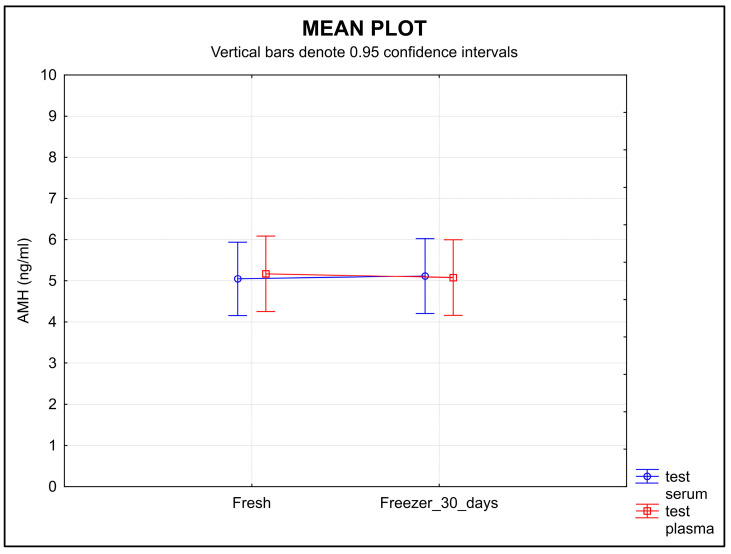
The stability of AMH during the 30-day storage at −20 °C. The molecule showed high stability in both serum and plasma samples. Legend: The stability of AMH in the samples stored for 30 days at −20 °C did not show any significant change in concentration over time in either type of sample. Equivalence was proved at a ±0.30 ng/mL limit of tolerance (*p* < 0.0001 for serum, *p* = 0.0238 for plasma).

**Table 1 diagnostics-13-01501-t001:** The protocol of plasma and serum sample stability testing.

Start Storage at	No of Aliquots	F/T Cycle	Final Storage at
	1	0	
	2	1	
−80 °C	3	2	−80 °C
	4	3	
	5	4	
	6	5	
**Start storage at**	**No of aliquots**	**Storage Period**	**Final storage at**
		1 h	
		2 h	
RT;	7–12;	4 h	−80 °C
4 °C	13–18	24 h	
		72 h	
		168 h	
−20 °C	19	30 d	−80 °C

No-number; F/T Cycle-thawing 2 h at RT and the subsequent storage of the sample for at least 12 h at −80 °C; RT—room temperature; h—hour; d—days.

## Data Availability

Data are available from the corresponding author.

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
