# Peer review of "The Stability of the Anti-Müllerian Hormone in Serum and Plasma Samples under Various Preanalytical Conditions"

_diagnostics, 2023, doi:10.3390/diagnostics13081501_

Round 1
Reviewer 1 Report
This is a well-written study worthy of publication after major revision according to the following comments.
- The main limitation of this study is small sample size; as the authors mention (lines 143-145), the two male subjects were excluded, leaving 19 blood samples from young women. Hence, statistical revision is needed, including power analysis; if power analysis proves that sample size was inadequate, further blood samples should be tested before publication.
- In lines 61-63, the authors assume that use of hormonal contraception methods had no influence on the study, since “they do not change the chemical structure and stability of AMH”. However, this is merely an assumption, not based on hard evidence. The authors should discuss this limitation in the Discussion section.
- In respect to the day of the menstrual cycle, it seems that blood samples were collected randomly. However, evidence exists showing that AMH levels vary significantly within the menstrual cycle [see Melado L, Lawrenz B, Sibal J, Abu E, Coughlan C, Navarro AT, Fatemi HM. Anti-müllerian Hormone During Natural Cycle Presents Significant Intra and Intercycle Variations When Measured With Fully Automated Assay. Front Endocrinol (Lausanne) 2018 Nov 27;9: 686. doi: 10.3389/fendo.2018.00686]. The authors should discuss this limitation in the Discussion section.
- In line 102 the authors state: “Statistical significance was determined at 0.05.” It is more appropriate to say that p-value <0.05 was considered to be statistical significant. Please rephrase accordingly.
Reviewer 2 Report
The authors conduct a study of serum and lithium plasma AMH stability using samples collected from females. Prior publications have also explored this topic.
Significant study limitations are the exclusion of samples from males, the small sample size and the short term duration of study.
1. Specify the time point(s) for serum separation within the protocol.
2. Provide the AMH assay sensitivity.
3. Increase the sample size, especially for the one month of freezer storage where only aliquot 19 was tested (n = 1).
4. The x axis of Figure 2 is not clear. Does C = cycles?
5. Explain exclusion of results from males in the Methods or Results, rather than in the Discussion.
6. The biochemical forms of AMH are postulated to differ in women with and without PCOS. It is important to consider this variable in the assessment of stability.
Round 2
Reviewer 2 Report
revised in accordance with suggestions